# Characterization of Phosphorylation Status and Kinase Activity of Src Family Kinases Expressed in Cell-Based and Cell-Free Protein Expression Systems

**DOI:** 10.3390/biom11101448

**Published:** 2021-10-02

**Authors:** Emiko Kinoshita-Kikuta, Eiji Kinoshita, Misaki Suga, Mana Higashida, Yuka Yamane, Tomoka Nakamura, Tohru Koike

**Affiliations:** 1Department of Functional Molecular Science, Graduate School of Biomedical and Health Sciences, Hiroshima University, Hiroshima 734-8553, Japan; tkoike@hiroshima-u.ac.jp; 2Department of Human Nutrition, Faculty of Human Sciences, Hiroshima Bunkyo University, Hiroshima 732-0295, Japan; kinoshitae@h-bunkyo.ac.jp; 3School of Pharmaceutical Science, Hiroshima University, Hiroshima 734-8553, Japan; b174399@hiroshima-u.ac.jp (M.S.); b172569@hiroshima-u.ac.jp (M.H.); b174670@hiroshima-u.ac.jp (Y.Y.); m210836@hiroshima-u.ac.jp (T.N.)

**Keywords:** Src family kinase, phosphorylation, protein expression system, phosphorylation, phos-tag

## Abstract

The production of heterologous proteins is an important procedure for biologists in basic and applied sciences. A variety of cell-based and cell-free protein expression systems are available to achieve this. The expression system must be selected carefully, especially for target proteins that require post-translational modifications. In this study, human Src family kinases were prepared using six different protein expression systems: 293 human embryonic kidney cells, *Escherichia coli,* and cell-free expression systems derived from rabbit reticulocytes, wheat germ, insect cells, or *Escherichia coli*. The phosphorylation status of each kinase was analyzed by Phos-tag SDS-PAGE. The kinase activities were also investigated. In the eukaryotic systems, multiple phosphorylated forms of the expressed kinases were observed. In the rabbit reticulocyte lysate system and 293 cells, differences in phosphorylation status between the wild-type and kinase-dead mutants were observed. Whether the expressed kinase was active depended on the properties of both the kinase and each expression system. In the prokaryotic systems, Src and Hck were expressed in autophosphorylated active forms. Clear differences in post-translational phosphorylation among the protein expression systems were revealed. These results provide useful information for preparing functional proteins regulated by phosphorylation.

## 1. Introduction

The field of proteomics involves investigating any aspect of a protein, such as its structure, modification, localization, and interactions with proteins or other molecules. To study the biological function of a specific protein, researchers select a certain method to produce the functional protein of interest. A variety of protein expression systems, including cell-based and cell-free ones, have been developed for this purpose.

*Escherichia coli* is one of the most convenient organisms for producing recombinant proteins [1]. *E. coli* protein expression systems do not usually enable post-translational modifications such as phosphorylation, glycosylation, and lipidation, so they are mainly suitable for preparing prokaryotic proteins or eukaryotic proteins that do not require post-translational modifications. Eukaryotic protein expression systems, such as baculovirus-insect cells, yeast, and mammalian cells, have the capacity to carry out post-translational modifications to express functional eukaryotic proteins, such as active kinases [2,3,4]. These cell-based systems have the advantage of a high yield, and the prepared proteins are also applicable for structural analysis, functional assays, the study of protein–protein interactions, and antibody production, among others. In addition to these research objectives, these cell-based expression systems are frequently employed for the production of recombinant proteins as therapeutics [4].

Cell-free expression systems, also known as in vitro transcription/translation systems, have become a popular and useful approach in basic and applied sciences [5,6,7,8,9]. Because cell-free systems have the ability to produce proteins in a rapid and high-throughput manner, they are increasingly being used in high-throughput functional genomics and proteomics. In recent years, propelled by the increasing production capacity, cell-free systems have attracted interest not only for research purposes but also in the pharmaceutical industry [5,10]. Using cell-free systems, protein synthesis can be performed in coupled transcription and translation systems using a DNA template within a few hours. The systems contain macromolecular components required for transcription and translation. The most frequently used cell-free expression systems have been developed based on a cell extract; for example, *E. coli*, rabbit reticulocytes, wheat germ, insect cells, and human cells [5,6,7,8]. These cells function in different ways, and each has its own advantages and disadvantages [6]. The choice of system is dependent on several factors, including the origin of the template RNA and DNA, the protein yield, and whether the protein of interest requires post-translational modification. In this study, to investigate the difference in post-translational modification of human tyrosine kinases, especially phosphorylation, among the cell-free expression systems, four T_N_T cell-free expression systems provided by Promega were used. These systems include different types of cell lysate: the T_N_T SP6 Quick Coupled Transcription/Translation System (a rabbit reticulocyte lysate system) [11,12], the T_N_T SP6 High-Yield Wheat Germ Protein Expression System [11,12], the T_N_T T7 Insect Cell Extract Protein Expression System [13], and the S30 T7 High-Yield Protein Expression System (an *E. coli* lysate system) [14,15,16].

The activity of cytoplasmic tyrosine kinases, the Src family kinases (SFKs; Src, Fgr, Hck, Yes, Blk, Fyn, Lck, and Lyn), is stringently regulated by their phosphorylation and dephosphorylation status in human cells [17]. As for Src, the SH2 domain interacts with phosphorylated Y527. Y527 in Src and the corresponding tyrosine in other SFKs are the primary sites of tyrosine phosphorylation in vivo (the sequence alignment of eight SFKs and tyrosine phosphorylation sites is shown in Appendix A). This residue is phosphorylated by the cytoplasmic tyrosine kinase Csk to keep the kinase in a closed, inactive conformation. The autophosphorylation site corresponds to Y416 in Src, and the corresponding tyrosine in other SFKs within the catalytic domain is also important for regulating kinase activity. Phosphorylation of this tyrosine allows the kinase to adopt an active conformation.

In previous studies, the protein phosphorylation status was analyzed by the phosphate affinity electrophoresis technique Phos-tag SDS-PAGE [18,19,20,21,22]. We found that multiple variants of SFKs with different phosphorylation statuses are constitutively present in human cells [23]. In this study, the SFKs expressed in cell-free expression systems were electrophoresed in Phos-tag gel and the banding patterns were compared with those of SFKs expressed in human cells. Moreover, the phosphorylation status and kinase activity of SFKs expressed in living *E. coli* were also analyzed using a method combining Phos-tag SDS-PAGE and the *E. coli* expression system, which we reported in our recent paper [24]. Using this method, SFKs and their common substrate, glutathione S-transferase-tagged Srctide (GST-Srctide, [25]), were co-expressed in *E. coli* and both tyrosine phosphorylation of GST-Srctide and the phosphorylation status of the kinases were analyzed.

## 2. Materials and Methods

### 2.1. Materials

Phos-tag Acrylamide AAL-107, Screen*F*ect A *plus,* anti-6His tag antibody, anti-FLAG antibody magnetic beads, and anti-GST antibody were purchased from Fujifilm Wako Pure Chemical Corp. (Osaka, Japan). The 293 (human embryonic kidney) cell line and cDNAs encoding human Src (clone IRAL047C19), Yes (clone W01A026C12), Fyn (clone W01A056G21), Lck (clone W01A107I15), Hck (clone IRAL034D12), Blk (clone W01A026C12), and Lyn (clone Lyn/pLY30) were obtained from RIKEN BioResource Research Center (RIKEN BRC, Tsukuba, Japan). cDNA encoding human Fgr (clone pF1KB9941), the T_N_T SP6 Quick Coupled Transcription/Translation System, the T_N_T T7 Insect Cell Extract Protein Expression System, the T_N_T SP6 High-Yield Wheat Germ Protein Expression System, the S30 T7 High-Yield Protein Expression System, pF25A ICE T7 Flexi Vector, and pSP64 poly(A) vector were purchased from Promega Corp. (Madison, WI, USA). Anti-phosphotyrosine antibody (PY20) was purchased from Santa Cruz Biotechnology (Dallas, TX, USA). Anti-FLAG antibody, ampicillin, streptomycin, and bovine intestinal mucosa alkaline phosphatase were obtained from Sigma-Aldrich (St. Louis, MO, USA). pQE30 vector was obtained from Qiagen (Hilden, Germany). pCDF1b vector and *E. coli* BL21(DE3) were purchased from Merck Millipore (Darmstadt, Germany). pGEX 6p-1 vector was purchased from Cytiva (Sheffield, UK). pHEK293 Ultra Expression Vector I and the In-Fusion HD Cloning Kit were purchased from Takara Bio Inc. (Kusatsu, Japan). pCDNA3.1(–) was obtained from Thermo Fisher Scientific (Carlsbad, CA, USA). GST-Accept resin and isopropyl-β-D-thiogalactoside (IPTG) were purchased from Nacalai Tesque (Kyoto, Japan). Human protein tyrosine phosphatase 1B (PTP 1B) was purchased from Abcam (Cambridge, UK).

### 2.2. Plasmid Construction

The plasmids shown in Figure 1 were constructed. Plasmid construction experiments were performed using the In-Fusion HD Cloning Kit. Two complementary DNA oligomers encoding Srctide (GEEPLYWSFPAKKK) [25] were annealed to form a double-stranded DNA sequence that was inserted between EcoRI and XhoI of pGEX-6p-1 to construct pGEX-6P-1_Srctide. In pGEX-6P-1_Srctide, the sequence of XhoI was replaced with tcgag (see the primer sequence of pGEX-6P-1_ΔXhoI in Appendix A). Therefore, the resulting construct was confirmed by digestion with EcoRI and by the lack of digestion with XhoI. The sequence of the GST-tagged Srctide was amplified from pGEX-6P-1_Srctide and inserted between BamHI and XhoI of pCDF-1b to construct pCDF_GST-Srctide. The UniProt Accession No. of each SFK and expression vectors for subcloning are summarized in Table 1. All DNA oligomers used for the plasmid construction are shown in Appendix A. For the construction of pCDNA3.1(−)_SFKs, each SFK sequence (Src, Fgr, Hck, Yes, Blk, Fyn, Lck, or Lyn) was designed and amplified to include a FLAG tag at the C-terminal end and inserted between EcoRI and BamHI of pcDNA3.1(−). For the construction of pHEK293 Ultra Expression Vector_SFKs, each SFK sequence was designed and amplified to include a FLAG tag at the C-terminal end and inserted between BamHI and XbaI of pHEK293 Ultra Expression Vector I. For the construction of pET21a(+)_SFKs, each amplified SFK sequence was inserted between BamHI and XhoI of pET21a(+). For the construction of pSP64 poly(A)_ SFKs, each SFK sequence was designed and amplified to include a FLAG tag at the C-terminal end and inserted between PstI and XbaI of pSP64 poly(A) vector. For the construction of pFN25A T7 ICE Flexi_SFKs, each SFK sequence was designed and amplified to include a FLAG tag at the C-terminal end and inserted between SgfI and PmeI of pFN25A T7 ICE Flexi Vector. For the construction of pQE30_SFKs, each SFK sequence was designed and amplified to include a FLAG tag at the C-terminal end and inserted between BamHI and HindIII of pQE30. Mutagenesis of SFKs was performed using the primers listed in Appendix A. All sequences into which mutations were introduced were confirmed using an ABI PRISM 3130XL Genetic Analyzer (Applied Biosystems, Foster City, CA, USA).

### 2.3. Transformation of Plasmids and Protein Expression

For co-transformation, a plasmid combination of pCDF_GST-Srctide and each pET21a(+)_SFK was co-transformed into the expression host strain, *E. coli* BL21(DE3). A single colony isolated on an LB agar plate containing ampicillin (100 µg/mL) and streptomycin (50 µg/mL) was transferred into 4 mL of LB medium. To induce co-expression of the cloned genes, IPTG (1 mM) was added to the medium during the early logarithmic-growth phase and incubated for 6 h. The collected cells were suspended in 500 µL of 10 mM Tris–HCl (pH 8.0) and sonicated to obtain a soluble fraction of lysate. For GST-Srctide purification, pGEX-6P-1 was transformed into the strain *E. coli* JM109. To induce expression of GST-Srctide, IPTG (1 mM) was added to the medium during the early logarithmic-growth phase and incubated for 6 h. GST-tagged proteins were purified using GST-Accept resin and stocked in 50 mM Tris–HCl (pH 8.0) containing 50% glycerol at –20 °C.

### 2.4. Phos-Tag SDS-PAGE

Phos-tag SDS-PAGE was performed using a 1-mm-thick, 9-cm-wide, and 9-cm-long gel on a mini-type PAGE apparatus (AE-6500; Atto Corp., Tokyo, Japan). We used a separating gel (6.3 mL) consisting of 7% or 10% *w*/*v* polyacrylamide and 357 mM 2-[bis(2-hydroxyethyl)amino]-2-(hydroxymethyl)propane-1,3-diol hydrochloride (Bis-Tris–HCl) buffer (pH 6.8), together with a stacking gel (1.8 mL) consisting of 4% *w*/*v* polyacrylamide and 357 mM Bis-Tris–HCl buffer (pH 6.8) as a neutral phosphate-affinity SDS-PAGE system [21]. Phos-tag Acrylamide AAL-107 (20 μM) and two equivalents of ZnCl_2_ (40 μM) were added to the separating gel before polymerization. An acrylamide stock solution was prepared to contain a 39:1 acrylamide–N,N′-methylenebisacrylamide mixture. The running buffer consisted of 0.10 M Tris and 0.10 M 3-morpholinopropane-1-sulfonic acid (MOPS) containing 0.10% *w*/*v* SDS and 5.0 mM NaHSO_3_, the latter being added immediately before use. Electrophoresis was performed at 30 mA/gel until the bromophenol blue dye reached the bottom of the separating gel. Subsequently, Western blotting was performed by the wet-tank method [26]. The Phos-tag SDS-PAGE images shown in this study were selected from images from three or more independent experiments showing a clear banding pattern without distortion. The raw images are shown in Appendix A.

### 2.5. Cell Culture and Transfection

The 293 cells were incubated in Dulbecco’s modified Eagle’s medium (DMEM) supplemented with 10% *v*/*v* fetal bovine serum, 100 units/mL penicillin, and 100 μg/mL streptomycin. For the analysis using Phos-tag SDS-PAGE, 1 × 10^5^ cells/100 µL were plated onto a 96-well plate for 6 h, and each expression vector (2 μg) was transfected into the cells with Screen*F*ect A *plus* reagents. After incubation for 20 h, the transfected cells were gently washed with Tris-buffered saline and immediately lysed with a sample-loading buffer for SDS-PAGE (65 mM Tris–HCl (pH 6.8), 1% *w*/*v* SDS, 10% *v*/*v* glycerol, 5% *v*/*v* 2-sulfanylethanol, and 0.03% *w*/*v* bromophenol blue) or a cell lysis buffer for immunoprecipitation (50 mM Tris–HCl (pH 8.0), 150 mM NaCl, 0.5%(*w*/*v*) sodium deoxycholate, 1.0%(*v*/*v*) Nonidet P40). Expression of the FLAG-tagged SFKs was confirmed by Western blotting with anti-FLAG antibody.

### 2.6. Cell-Free Protein Expression

For the T_N_T SP6 Quick Coupled Transcription/Translation System (the rabbit reticulocyte lysate system), each SFK gene was subcloned into pSP64 poly(A) Vector. Transcription and translation reactions were performed in a mixture composed of 4.0 µL of the T_N_T Quick Master Mix, 0.1 µL of 1 mM methionine, and 0.9 µL of each plasmid (0.09 µg) at 30 °C for 2 h. For the T_N_T T7 Insect Cell Extract Protein Expression System (the insect cell extract system), each SFK gene was subcloned into pF25A ICE T7 Flexi Vector. Transcription and translation reactions were performed in a mixture composed of 4.0 µL of the T_N_T T7 Insect Cell Extract Master Mix and 1.0 µL of each plasmid (0.1 µg) at 30 °C for 4 h. For the T_N_T SP6 High-Yield Wheat Germ Protein Expression System, each SFK gene was subcloned into pSP64 poly(A) Vector. Transcription and translation reactions were performed in a mixture composed of 3.0 µL of the T_N_T SP6 High-Yield Wheat Germ Master Mix and 2.0 µL of each plasmid (0.2 µg) at 25 °C for 2 h. For the S30 T7 High-Yield Protein Expression System (the *E. coli* lysate system), each SFK gene was subcloned into pQE30 Vector. Transcription and translation reactions were performed in a mixture composed of 1.8 µL of the T7 S30 Extract Circular, 2.0 µL of S30 Premix, and 1.2 µL of each plasmid (0.12 µg) at 37 °C for 1 h.

### 2.7. In Vitro Kinase Assay

Anti-FLAG antibody magnetic beads were equilibrated with cell lysis buffer. The 293 cells (5 × 10^5^ cells) expressing a FLAG-tagged kinase were washed with Tris-buffered saline and lysed in 100 µL of cell lysis buffer. A soluble fraction of the lysate or a cell-free protein expression reaction mixture was incubated with anti-FLAG antibody magnetic beads at 4 °C for 16 h. The beads were collected and washed with cell lysis buffer. The kinase assay was performed in 20 mM MOPS (pH 7.2) containing 25 mM β-glycerol phosphate, 1 mM Na_3_VO_4_, 30 mM MgCl_2_, 2 mM dithiothreitol, 2 mM ATP, magnetic beads, and 0.05 µg of GST-Srctide at 30 °C for 0–60 min. Reactions were terminated by adding a half volume of sample-loading buffer for SDS-PAGE.

## 3. Results

### 3.1. Kinase Activity and Phosphorylation Status of Human SFKs Expressed in 293 Cells

Since the phosphorylation status of human proteins expressed in human cell lines would reflect that of the proteins in a human cell, 293 human embryonic kidney cells were initially used for the SFK expression. The 293 cells were transfected with pcDNA3.1(−)_SFKs (Src, Lck, Hck, Blk, and the corresponding kinase-dead mutants) or pHEK293 Ultra Expression Vector_SFKs (Yes, Fyn, Lyn, Fgr, and the corresponding kinase-dead mutants) (see Figure 1 and Table 1). Kinase-dead mutants were prepared by replacing a tyrosine with a phenylalanine in the catalytic domain. The autophosphorylation of the tyrosine residue is important for adopting an active conformation [17]. The phosphorylation status of the wild-type (WT) SFKs was compared with the corresponding kinase-dead (KD) mutants by Phos-tag SDS-PAGE, followed by Western blotting with anti-FLAG antibody (Figure 2A). The open arrowhead shows the position of the unphosphorylated form, which was assigned by alkaline phosphatase treatment experiments (Appendix A). Phosphorylated forms were observed as multiple upshifted electrophoretic bands in all wild-type kinases. As for the kinase-dead mutants of Src, Hck, Yes, Blk, and Lyn, the number of phosphorylated forms decreased compared with that of the corresponding wild type.

Next, each expressed kinase was purified by immunoprecipitation with anti-FLAG antibody-bound magnetic beads and then subjected to an in vitro phosphorylation assay with GST-Srctide (GEEPLYWSFPAKKK tagged with GST, 28 kDa). The phosphorylation of GST-Srctide was confirmed using Phos-tag SDS-PAGE, followed by Western blotting with an anti-GST antibody (Figure 2B). An upshifted band corresponding to a phosphorylated form of GST-Srctide was observed in the lanes of Src, Hck, and Lyn, indicating that these three SFKs were constitutively activated in 293 cells. On the other hand, Fgr, Yes, Blk, Fyn, and Lck showed no phosphorylation activity. The kinase-dead mutant of Hck demonstrated lower but detectable activity compared with the wild type. The electrophoresis results using 293 cells were then compared with those of the other protein expression systems, as described in the following sections.

### 3.2. Kinase Activity and Phosphorylation Status of Human SFKs Expressed in E. coli

*E. coli* BL21(DE3) was co-transformed with pET21a(+)_SFKs (Src, Fgr, Hck, Yes, Blk, Fyn, Lck, Lyn, or the corresponding kinase-dead mutants) and pCDF_GST-Srctide (see Figure 1). The expression of 6His-tagged SFKs was confirmed by analysis using SDS-PAGE, followed by Western blotting with an anti-6His antibody (Figure 3A). The expression and phosphorylation of GST-Srctide were confirmed using Phos-tag SDS-PAGE, followed by Western blotting with an anti-GST antibody (Figure 3B). An upshifted band corresponding to a phosphorylated form of GST-Srctide was observed in each lane of Src (WT), Hck (WT), Yes (WT), and Lyn (WT), whereas no phosphorylated GST-Srctide was detected in the lanes of Fgr (WT), Blk (WT), Fyn (WT), Lck (WT), all kinase-dead mutants, and the control (pCDF_GST-Srctide alone). To confirm the phosphorylation of GST-Srctide, Src (WT) lysate was treated with alkaline phosphatase or tyrosine phosphatase (Appendix A). The phosphatase treatment returned the upshifted band to the unphosphorylated position. In the *E. coli* protein expression system, the fourth kinase, Yes, showed tyrosine-kinase activity (Figure 3B), in addition to Src, Hck, and Lyn, which were active in 293 cells (see Figure 2B). 

To verify the phosphorylation status of expressed SFKs and the corresponding kinase-dead mutants, the lysates were analyzed by Phos-tag SDS-PAGE, followed by Western blotting with 6His antibody (Figure 3C). Upshifted bands were observed in the lanes of Src (WT), Hck (WT), Yes (WT), and Lyn (WT), whereas no band in the upper region of the gel was detected in the lanes of Fgr (WT), Blk (WT), Fyn (WT), Lck (WT), and all kinase-dead mutants. These results are consistent with the phosphorylation activities of the four kinases Src, Hck, Yes, and Lyn toward GST-Srctide (see Figure 3B). In addition, each lysate was also analyzed by conventional SDS-PAGE followed by Western blotting with anti-pTyr antibody (Figure 3D). Tyrosine phosphorylation of endogenous proteins was detected in cells expressing the four kinases Src (WT), Hck (WT), Yes (WT), and Lyn (WT). 

### 3.3. Kinase Activity and Phosphorylation Status of SFKs Produced by Cell-Free Protein Expression Systems

Each SFK was expressed by the four different cell-free protein expression systems: the T_N_T SP6 Quick Coupled Transcription/Translation System (a rabbit reticulocyte lysate system), the T_N_T T7 Insect Cell Extract Protein Expression System, the T_N_T SP6 High-Yield Wheat Germ Protein Expression System, and the S30 T7 High-Yield Protein Expression System (an *E. coli* lysate system). The production of each kinase was confirmed by Western blotting with an anti-FLAG antibody (Figure 4A). The SFKs expressed in 293 cells (see Section 3.2 and Appendix A) were adopted as a reference sample (Lane H in Figure 4A). Slightly different mobilities on the conventional SDS-PAGE gel were observed for the same kinases, which would have been due to differences in post-translational modifications. The small molecule bands observed in some lanes are considered to be incomplete-length products.

Each expressed kinase was purified by immunoprecipitation with anti-FLAG antibody-bound magnetic beads and then subjected to an in vitro GST-Srctide phosphorylation assay. Each reaction was performed with an incubation time of 0, 10, 30, or 60 min, applied in order from the left of a Phos-tag gel (Figure 4B). Kinase activity was identified by an upshifted band assigned to the phosphorylated GST-Srctide on the Phos-tag SDS-PAGE. Src and Hck from all systems, Yes from the rabbit reticulocyte, insect cell, and wheat germ systems, Lyn from the rabbit reticulocyte and insect cell systems, and Fgr and Blk from the rabbit reticulocyte system were active in GST-Srctide phosphorylation, whereas Fyn and Lck expressed in all systems showed no activity.

To visualize the difference in the phosphorylation status among the protein expression systems, the SFK produced by each system was electrophoresed in a Phos-tag gel (Figure 4C). The open arrowheads in this figure show the positions of the unphosphorylated forms, which were assigned by the banding pattern of the SFK expressed in 293 cells (lane H). In comparison with the electrophoresis bands on the conventional SDS-PAGE (Figure 4A), all SFKs from rabbit reticulocytes (lane R), insect cells (lane I), wheat germ (lane W), and 293 cells (lane H) clearly showed multiple upshifted bands, indicating post-translational phosphorylation in the expression systems. In the *E. coli* expression system (lane E), no upshifted bands were observed for Fgr, Yes, Fyn, Lck, and Lyn, indicating that *E. coli* protein expression systems do not usually enable post-translational phosphorylation. The upshifted bands in Src, Hck, and Blk expressed in the *E. coli* expression system might have been due to autophosphorylation. The Phos-tag banding patterns were unique to each protein expression system. In addition, the ratio of the phosphorylated form to the unphosphorylated form depended on the properties of each kinase.

### 3.4. Phosphorylation Status of Kinase-Dead Mutants Expressed in Cell-Free Systems

Kinase-dead mutants were expressed in the cell-free systems and the effects of the mutation on the kinase activities and phosphorylation status of SFKs were investigated. First, we confirmed that all mutants were inactive in phosphorylation assays using GST-Srctide (Appendix A). Next, the Phos-tag banding patterns were compared between the wild type and the corresponding kinase-dead mutants (Figure 5). In the rabbit reticulocyte system, Src, Fgr, Hck, Yes, Blk, and Lyn showed significantly different patterns between the wild type and the kinase-dead mutant; some of the phosphorylation bands observed in the wild type disappeared in the kinase-dead mutant. In contrast, the banding patterns of each kinase-dead mutant from the other three systems were almost the same as those of the corresponding wild types. According to the phosphoproteomics database PhosphoSitePlus (https://www.phosphosite.org/homeAction.action, accessed on 1 October 2021), multiple phosphorylation sites for each SFK have been reported: 32 for Src, 15 for Fgr, 19 for HCK, 30 for Yes, 14 for Blk, 36 for Lck, 22 for Fyn, and 38 for Lyn. Both wild types and kinase-dead mutants would be phosphorylated at multiple sites other than the autophosphorylation site and they showed complicated band patterns. The multiple upshifted bands observed in each kinase produced by the four cell-free systems return to their unphosphorylated position by alkaline phosphatase treatment (see Appendix A), so upshifted bands were due to phosphorylation and not other modifications. As shown in Figure 2A, Src, Fgr, Hck, Yes, Blk, and Lyn expressed in 293 cells also showed different banding patterns between the wild type and the kinase-dead mutant. This indicates that the rabbit reticulocyte system would have similarities to human cells in terms of the mechanism of post-translational phosphorylation.

## 4. Discussion

In this study, we prepared eight human SFKs by six different protein expression methods using 293 human embryonic kidney cells, an *E. coli* protein expression system, and four types of cell-free protein expression system. The phosphorylation status of the expressed SFKs was analyzed using Phos-tag SDS-PAGE and the correlation of phosphorylation status and GST-Srctide phosphorylation activity was investigated.

In 293 human embryonic kidney cells, multiple forms of SFKs in different phosphorylation statuses were identified (Figure 2A). Some of these phosphorylated forms disappeared in the kinase-dead mutants, which lack the autophosphorylation site required for adopting an active conformation. Remarkable differences between the wild types and the kinase-dead mutants were observed for Src, Hck, Yes, Blk, and Lyn, indicating that those active species are constitutively expressed in 293 cells. Src, Hck, and Lyn showed GST-Srctide phosphorylation activity, whereas the other SFKs showed no such activity (Figure 2B). Contrary to expectations, the Hck kinase-dead mutant had weak but detectable activity, and Yes and Blk showed no activity. These results suggest that Hck, Yes, and Blk are regulated by other mechanisms in addition to the autophosphorylation of the catalytic domain, under the experimental conditions. It has been reported that more than one mechanism is often involved in SFK activation in response to a single stimulus [17]. For example, in B cells, CSK and a certain protein tyrosine phosphatase(s) are balanced to maintain Lyn, Fyn, Lck, and Blk activation in response to stimulation of the B-cell antigen receptor [27]. Lck and Fyn, which were not active in 293 cells, are involved in T-cell antigen receptor signaling. It has been reported that they must each be recruited to the stimulated T-cell antigen receptor complex and then activated [28].

The *E. coli* protein expression system is a convenient tool, but large-scale expression of eukaryotic protein kinases using it is not always successful. For tyrosine protein kinases in particular, problems such as low expression or the formation of inclusion bodies are frequently encountered. This might be due to the potential cytotoxicity of the tyrosine protein kinases or to misfolding [29,30]. Although large-scale expression of a human tyrosine kinase in *E. coli* is difficult, we have confirmed that all SFKs can be expressed at levels detectable by Western blotting (Figure 3A). We showed that the SFKs and their substrate (GST-Srctide) could be co-expressed in *E. coli* and that the GST-Srctide was phosphorylated in *E. coli* at the tyrosine residue by Src, Hck, Yes, and Lyn (Figure 3B). These four kinases were also phosphorylated themselves (Figure 3C). Since there is no endogenous tyrosine kinase in *E. coli*, it is considered that this was due to autophosphorylation. The phosphorylated forms would contain active forms involved in phosphorylation of the GST-Srctide and *E. coli* endogenous proteins (Figure 3D). Non-specific phosphorylation of cellular proteins suggests that the functional heterologous kinase expression would alter the original functions of certain endogenous proteins. This may be one of the reasons for the difficulty of preparing human tyrosine kinases using *E. coli* hosts, as the toxicity of heterologous proteins in *E. coli* has been pointed out [30].

Post-translational phosphorylation may be the most important factor to consider when preparing active SFKs using cell-free protein expression systems. We have thus been interested in the differences in phosphorylation among different types of cell-free system and their ability to produce functional kinases. Src and Hck synthesized in all systems showed GST-Srctide phosphorylation activity (Figure 4B). Since the *E. coli* cell extract lacks post-translational phosphorylation, Src and Hck would be activated by autophosphorylation. The autophosphorylation of Src and Hck was also observed in *E. coli* BL21 (DE3) (Figure 3C). Yes and Lyn from the *E. coli* cell-free system were inactive in GST-Srctide phosphorylation (Figure 4B), while *E. coli* BL21 (DE3) provided active forms of Yes and Lyn (Figure 3B,C). This difference indicates that *E. coli* cells contain additional factors required for the autophosphorylation of Yes and Lyn. Meanwhile, Fyn and Lck were inactive in all systems (Figure 4B), suggesting that they are strictly regulated by specific mechanisms in human cells [28]. For Fgr and Blk, active kinases were produced only in the rabbit reticulocyte system. The Phos-tag SDS-PAGE of each kinase (Figure 4C) showed the difference in phosphorylation status between the cell-free systems. No clear correlation could be found between the Phos-tag banding pattern and the kinase activity; however, it became clear that there were significant differences in the post-translational phosphorylation mechanism among the systems.

Table 2 summarizes the presence (+) or absence (−) of the activity of SFKs expressed in the four cell-free systems together with the results of 293 cells (see Section 3.1) and *E. coli* BL21 (DE3) (see Section 3.2). The rabbit reticulocyte, insect cell, wheat germ, and *E. coli* systems produced six, five, three, and two active kinases, respectively. A characteristic difference between the rabbit reticulocyte system, which produced the largest number of active kinases, and the other three systems was found in the Phos-tag pattern (Figure 5). Only in the rabbit reticulocyte system did expressed kinases show a difference in phosphorylation status between the wild type and the kinase-dead mutant. The kinases expressed in 293 cells also showed differences in Phos-tag pattern between the wild type and the kinase-dead mutant (Figure 2A), which was similar to the characteristics of the kinases expressed in the rabbit reticulocyte system. This indicates that the rabbit reticulocyte system and human cells would have similarities in terms of the mechanism of post-translational phosphorylation, for example, in the types of functioning kinases [6]. The mammalian-derived rabbit reticulocyte system may be more suitable than the other three systems for producing active human kinases.

When selecting protein expression systems for individual research, post-translational modifications should be taken into consideration. Rabbit reticulocyte and insect cell systems have shown the most versatility for various modifications: isoprenylation [31,32], acetylation [33,34], N-myristoylation [34], phosphorylation [35], ubiquitination [36], and glycosylation [37,38]. The wheat germ system is not suitable for glycosylation [38], whereas it is the most efficient at making a wide variety of proteins and sophisticated for high-throughput methods for functional genomics and proteomics [9,39]. This study demonstrates that the phosphorylation mechanism differed significantly among cell-free systems, as well as between cell-free systems and actual human cells. Moreover, we showed that the activation of produced SFKs depends on both the characteristics of the protein and the cell-free system. These results provide useful information for proteomics approaches, especially for exploring functional proteins requiring post-translational modifications.

## Figures and Tables

**Figure 1 biomolecules-11-01448-f001:**
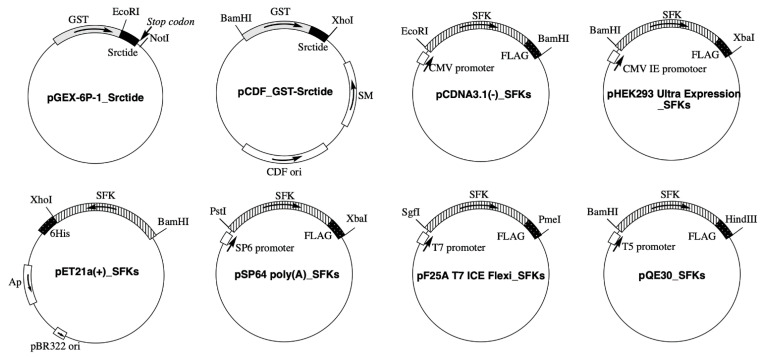
The plasmids constructed in this study.

**Figure 2 biomolecules-11-01448-f002:**
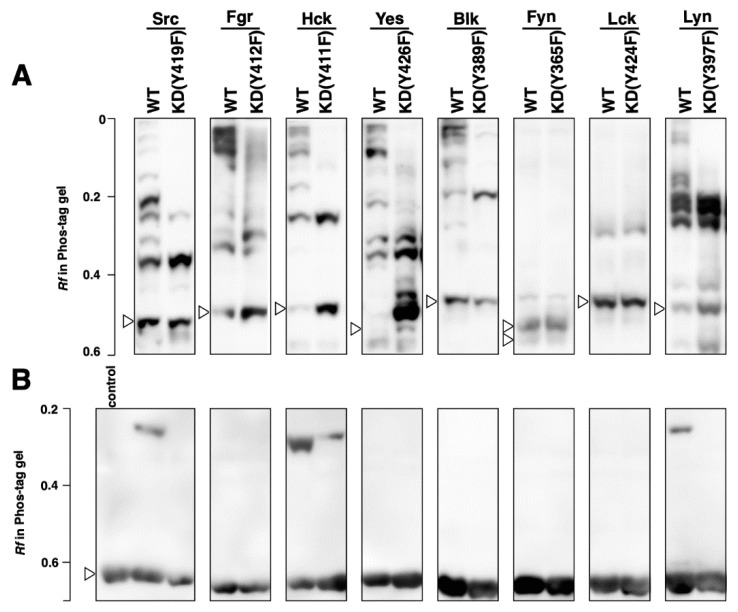
Phosphorylation status and tyrosine kinase activity of SFKs expressed in 293 cells. (**A**) Phosphorylation status of the wild-type (WT) SFKs and the corresponding kinase-dead (KD) mutants expressed in 293 cells were analyzed by Phos-tag SDS-PAGE (7% *w*/*v* polyacrylamide, 20 µM Zn^2+^–Phos-tag), followed by Western blotting with anti-FLAG antibody. The open arrowheads show the positions of the unphosphorylated forms. (**B**) The SFKs expressed in 293 cells were purified by immunoprecipitation with anti-FLAG antibody-bound magnetic beads and then subjected to an in vitro GST-Srctide phosphorylation assay. Each reaction was performed for 60 min. Reactions were analyzed by Phos-tag SDS-PAGE (10% *w*/*v* polyacrylamide, 20 µM Zn^2+^–Phos-tag), followed by Western blotting with anti-GST antibody. The open arrowhead shows the positions of the unphosphorylated GST-Srctides.

**Figure 3 biomolecules-11-01448-f003:**
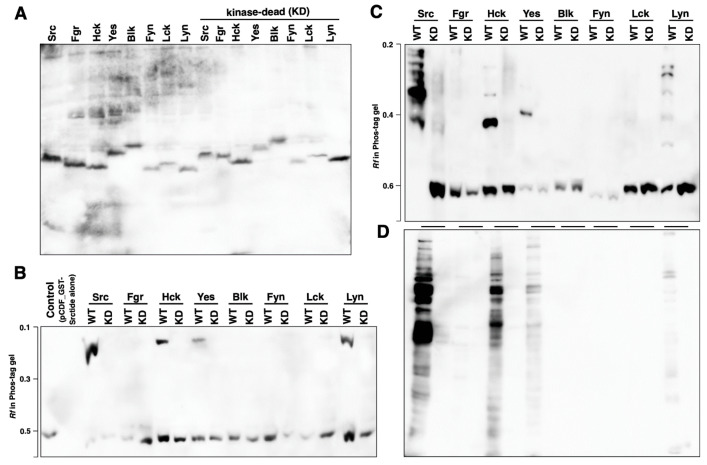
Tyrosine kinase activity and phosphorylation status of SFKs expressed in *E. coli*. (**A**) Expression of the wild-type (WT) SFKs and the corresponding kinase-dead (KD) mutants in *E. coli* BL21(DE3) co-transformed with pET21a(+)_SFKs and pCDF_GST-Srctide was confirmed by Western blotting with anti-6His antibody. (**B**) Expression and phosphorylation of GST-Srctide in co-transformed *E. coli* were confirmed by Phos-tag SDS-PAGE (10% *w*/*v* polyacrylamide, 20 µM Zn^2+^–Phos-tag), followed by Western blotting with anti-GST antibody. (**C**) Phosphorylation status of SFKs in co-transformed *E. coli* was analyzed by Phos-tag SDS-PAGE (7% *w*/*v* polyacrylamide, 20 µM Zn^2+^–Phos-tag), followed by Western blotting with anti-6His antibody. (**D**) Phosphorylation of endogenous proteins of co-transformed *E. coli* was analyzed by Western blotting with anti-phosphotyrosine antibody.

**Figure 4 biomolecules-11-01448-f004:**
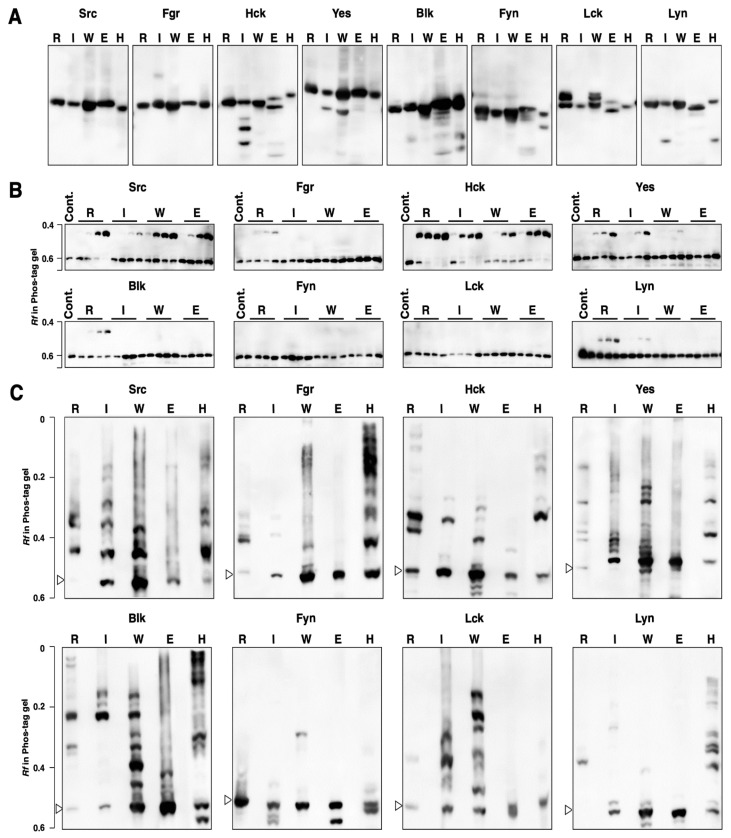
Phosphorylation status and tyrosine kinase activity of SFKs produced by cell-free protein expression systems. (**A**) Expression of SFKs by the T_N_T SP6 Quick Coupled Transcription/Translation System (a rabbit reticulocyte lysate system, lane R), the T_N_T T7 Insect Cell Extract Protein Expression System (lane I), the T_N_T SP6 High-Yield Wheat Germ Protein Expression System (lane W), the S30 T7 High-Yield Protein Expression System (an *E. coli* lysate system, lane E), and 293 cells (lane H) was confirmed by Western blotting with anti-FLAG antibody. (**B**) SFKs expressed by each cell-free protein expression system were purified by immunoprecipitation with anti-FLAG antibody-bound magnetic beads and then subjected to an in vitro GST-Srctide phosphorylation assay. For each reaction, 0, 10, 30, and 60 min reactions were applied in order from the left in Phos-tag SDS-PAGE gels (10% *w*/*v* polyacrylamide, 20 µM Zn^2+^–Phos-tag). The gels were analyzed by Western blotting with anti-GST antibody. (**C**) Phosphorylation status of SFKs expressed in cell-free protein expression systems and in 293 cells was analyzed by Phos-tag SDS-PAGE (7% *w*/*v* polyacrylamide, 20 µM Zn^2+^–Phos-tag), followed by Western blotting with anti-FLAG antibody. The open arrowheads show the positions of the unphosphorylated forms.

**Figure 5 biomolecules-11-01448-f005:**
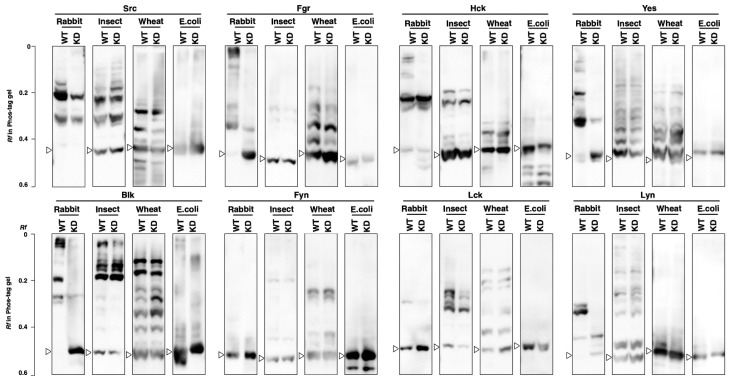
Phosphorylation status of the wild-type (WT) SFKs and the corresponding kinase-dead (KD) mutants produced by the T_N_T SP6 Quick Coupled Transcription/Translation System (a rabbit reticulocyte lysate system), the T_N_T T7 Insect Cell Extract Protein Expression System, the T_N_T SP6 High-Yield Wheat Germ Protein Expression System), and the S30 T7 High-Yield Protein Expression System (*E. coli* lysate system) was analyzed by Phos-tag SDS-PAGE (7% *w*/*v* polyacrylamide, 20 µM Zn^2+^–Phos-tag), followed by Western blotting with anti-FLAG antibody. The open arrowheads show the positions of the unphosphorylated forms.

**Table 1 biomolecules-11-01448-t001:** The Uniprot Accession No. of each SFK and expression vector for subcloning.

SFK	UniProt Accession No.	Expression Vector				
		*E. coli* BL21(DE3)	293 Cells	For Cell-Free Protein Expression System		
				Rabbit Reticulocytes, Wheat Germ	Insect Cells	*E. coli*
Src	P12931	pET21a(+)	pcDNA3.1(−)	pSP64 poly(A) Vector	pF25A ICE T7 Flexi Vector	pQE30
Lck	P06239					
Hck	P08631					
Blk	P51451					
Yes	P07947		pHEK293 Ultra Expression Vector I			
Fyn	P06241					
Lyn	P07948					
Fgr	P09769					

**Table 2 biomolecules-11-01448-t002:** Summary of the kinase activity of SFKs expressed in cell-free protein expression systems, 293 cells, and *E. coli* BL21(DE3). + or − means presence or absence of phosphorylation activity toward GST-Srctide, respectively.

SFK	293 Cells	*E. coli* BL21(DE3)	Cell-Free Protein Expression System			
			Rabbit Reticulocytes	Insect Cells	Wheat Germ	*E. coli*
Src	+	+	+	+	+	+
Fgr	−	−	+	−	−	−
Hck	+	+	+	+	+	+
Yes	−	+	+	+	+	−
Blk	−	−	+	+	−	−
Fyn	−	−	−	−	−	−
Lck	−	−	−	−	−	−
Lyn	+	+	+	+	−	−

## Data Availability

Not applicable.

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
