# Peer review of "Characterization of Phosphorylation Status and Kinase Activity of Src Family Kinases Expressed in Cell-Based and Cell-Free Protein Expression Systems"

_biomolecules, 2021, doi:10.3390/biom11101448_

Round 1
Reviewer 1 Report
The authors of the manuscript entitled “Characterization of phosphorylation status and kinase activity of SRC tyrosine kinase family expressed in cell-based and cell-free protein expression systems” compared different cell-based and cell-free protein expression systems available and greatly used for the biologist community to produce heterologous proteins.
The authors compare the phosphorylation status of different kinases by Phostag SDS-PAGE electrophoresis and the kinase activities of the purified proteins by kinase assays and compared them with kinase-dead mutants. The results were well performed and raised interesting conclusions demonstrating that not all the systems to produce heterologous proteins render active proteins. The 293 cells and the rabbit reticulocyte lysate system seem to be the ones with better results. However, the authors did not discuss about why the rabbit reticulocyte system is better than the other cell-free system and this might be added in the discussion.
On the other hand, the other cell-free systems frequently gave several slow migrating bands in phostag gels. However, these band are similar in the kinase-dead mutants. Therefore, the authors suggest that they might correspond to other post-translational modifications. Did the authors treat the proteins with phosphatases to check whether the slow migrations bands are phosphorylations? Have been associated other post-translational modifications to be separated by Phostag gels? This is an important issue, since the Phostag gels are described to separate phosphorylations and widely used for this purpose. If they also separate another kind of post-translational modifications the Phostag should be put into question.
Author Response
Correspondence to Reviewer 1:
We really appreciate your constructive comments. The following are our replies to the comments.
Our response to major comment 1
Since the rabbit system is derived from the same mammalian cells as humans, it may only be best suited to produce active human kinases.
As our reply, we added the following sentences in Discussion (lines 456-458 ).
“The mammalian-derived rabbit reticulocyte system may be more suitable than the other three systems for producing active human kinases.”
Our response to major comment 2:
We believe that the shift-up band is due to phosphorylation. We have analyzed many proteins since we developed Phos-tag SDS-PAGE in 2006, and we have never observed such shift-up bands due to modifications other than phosphorylation.
As our reply, the supplemental Figure 6 has been added to show that the multiple bands observed in the proteins produced by the cell-free systems returned to their unphosphorylated position by AP treatment. And we added the following sentences in Section 3.4 (lines 371-374 ).
“The multiple upshifted bands observed in each kinase produced by the four cell-free systems return to their unphosphorylated position by alkaline phosphatase treatment (see Fig. S6), so upshifted bands were due to phosphorylation and not other modifications.”
Reviewer 2 Report
The Src-family protein tyrosine kinases (SFKs) are a group of non-receptor protein tyrosine kinases that play an essential role in the transmission of various extracellular signals. They are ubiquitously expressed in various cell types where they respond to activated GPCRs and other membranal receptors to induce a large number of stimulated process such as proliferation, differentiation migration and even apoptosis. No less than eight similar protein kinases have been shown as members of the group (Src, Fgr, Hck, Yes, Blk, Fyn, Lck, and Lyn) and they are all regulated by an incorporation of phosphates to either C-terminal tyrosine (inhibiting, Y527 in Src) or to a tyrosine in the active pocket of the kinase (activating, Y416 in Src). Although much information has already been accumulated on the SFKs, less is known on their structure-function relationships. Therefore, proper expression of the full length proteins is essential for characterization of the kinases’ properties and activities. In the current study the authors purified all eight SFKs in six different protein expression systems, examining their Tyr phosphorylation as well as activity that were compared to those of kinase dead mutants of the proteins. Clear differences in phosphorylation and activity among the expression systems were shown, providing information on the preparation of functional SFKs.
Although the authors undertaken a wide study that provides information on no less than 16 constructs and six distinct expression system, there are several major concerns with this study.
- The scientific advance of the study is not clear. The expression techniques, as well the activity assays have been used in the past to express protein kinases, and the authors do not improve or modulate any of them.
- The authors do not use their systems to study importnt structure-function relationships properties of the kinases, such is the rate of autophosphorylation, phosphorylation by c-CSK, myristylation or any other parameter.
- The comparison between the expression systems is not complete because the author do not provide information on the specific activity (activity per amount of proteins) and specific phosphorylation (phosphorylation per amount of protein) of the proteins. In principle it is possible that the parameter that is different between the systems is the yield of extraction or protein concertation, and not the actual activity of the distinct proteins.
Minor comments:
- It is suggested to use the term SFK (Src-family protein tyrosine kinases) and not SKF.
- In Fig. 2A, it is not clear what is the nature of the bands that are phosphorylated in the KD mutants.
Some of them seem to be stronger than in the WT proteins.
- There is no comparison of the WT-proteins to the endogenous ones. Tagged proteins may cause erroneous properties due to tag interference or mis-localization.
- The quality of the results in Fig. 3 can be improved.
- It is not clear what is the reason that some of the proteins (Fyn, Lck) are not active in any system. Can the author express them in T cells?
- In Fig. 4A, the nature of the lower molecular weight bands in some of the proteins/systems should be discussed. Are these proteins more susceptible to degradation?
Author Response
Correspondence to Reviewer 2:
We really appreciate your constructive comments. The following are our replies to the comments.
Our response to major comment 1:
The purpose of this study was not to create, improve, or modify cell-free systems, but to investigate the characteristics of existing systems. As mentioned in the Abstract, our objective is to show that there are differences in phosphorylation among cell-free systems and to guide the choice of a system in preparing the proteins that phosphorylation requires. Our data are the first demonstration to visualize the differences in the phosphorylation status and the activity among cell-free systems. The data provides a guidepost for choosing which cell-free system is best to use when preparing a target protein. It is also important to note that the wheat germ system has been used for the proteomic approach, but not all proteins produced by the system show in vivo function. As mentioned at the end of the Discussion (lines .469-471), we believe that the data provide important information for future proteomics studies.
Our response to major comment 2:
In this study, the rate of autophosphorylation and the phosphorylation by c-CSK were not considered. Bands derived from autophosphorylation can be identified by WB using the site-specific phospho-antibody, and the rate of autophosphorylation can be analyzed by quantifying the band density with reaction time. Similarly, phosphorylation by c-CSK can be identified by WB. However, in the cell-free systems, all SFKs are phosphorylated at multiple sites in addition to the autophosphorylation site or the site by c-CSK and show a complicated band pattern, so the analysis will be limited to a qualitative one. Quantitative analysis including kinetics is our future challenge.
 Myristoylation of SFKs is involved in localization to the cell membrane. Several studies have been reported about the correlation between cell membrane localization and kinase activity (for example, Patwardhan P. et al MCB, 2010,4094-4107, Myristoylation and membrane binding regulate c-Src stability and kinase activity). Since there is no organelle in the cell-free systems, the correlation between myristoylation and kinase activity was not considered in this study. Myristoylation occurs in both the insect cell system and rabbit reticulocyte one (ref.34) whereas there is no information about other systems. The correlation between myristoylation and kinase activity can be investigated by comparing the kinase activity between WT and G2A mutant of SFKs produced by the insect cell or the rabbit reticulocyte system. This is our future work.
Our response to major comment 3:
In the E.coli co-expression system (Fig. 3B), the expressed SFKs and the substrate GST-srctide were so small detectable by WB, and it was difficult to quantify the amount of protein and specific activity (activity per amount of proteins). Therefore, the measurement of using the co-expression system is not quantitative so we didn’t mention the comparison of activity among samples. However, the co-expression method has the advantage to easily determine whether the expressed kinase is active or not without purifying them.
Similarly, in vitro GST-srctide phosphorylation assay (Fig. 2B and Fig.4B) was performed to determine whether the expressed kinases are active or not. This experiment was not quantitative because it involves factors such as protein yield or efficiency of IP. So we couldn’t mention the comparison of activity among samples. However, this method has the advantage to easily determine whether the expressed kinase is active or not.
Our response to minor comment 1:
We replaced “src kinase family members (SKFs) “with “src family kinases (SFKs)”.
Our response to minor comment 2:
According to phosphoproteomics database, PhosphoSitePlus, multiple sites have been reported for SFKs that would be phosphorylated. Kinase-dead mutants would be phosphorylated at multiple sites other than the autophosphorylation site and they showed complicated band patterns. In point mutants such as kinase-dead, the density of each band often differs from that of WT. Therefore, only qualitative analysis is possible to discuss the presence or absence of the band. As our reply, we added the following sentences in section 3.4 (lines 366-371).
“According to phosphoproteomics database, PhosphoSitePlus (https://www.phosphosite.org/homeAction.action), multiple phosphorylation sites for each SFK have been reported; 32 for Src, 15 for Fgr, 19 for HCK, 30 for Yes, 14 for Blk, 36 for Lck, 22 for Fyn, and 38 for Lyn. Both wild types and kinase-dead mutants would be phosphorylated at multiple sites other than the autophosphorylation site and they showed complicated band patterns.”
Our response to minor comment 3:
We have previously reported that FLAG-tagged Lyn expressed in 293 cells is properly trafficked and localized to the cell membrane by immunofluorescence (ref.23). We have investigated only Lyn in detail, the expression of endogenous Lyn was low in 293 cells, and it could not be detected even by a reliable antibody (anti-Lyn (5G2) Mouse mAb available from CST). We believe that the expression vector used in this study allows FLAG-tagged SFKs to express much more compared with the expression level of endogenous ones.
Our response to minor comment 4:
As suggested by the reviewer, exactly, the S/N of Fig. 3A was high, but it was difficult to eliminate the non-specific signal because the target bands disappeared when the contrast was changed.
Our response to minor comment 5:
We consider that Lck and Fyn are activated in T-cells rather than 293. We didn’t have T-cell-derived cells and couldn’t try it. As our reply, Ref.28 is newly cited, and the following sentence is added in Discussion (lines 406-408).
“Lck and Fyn, which were not active in 293 cells, are involved in T-cell antigen receptor signaling. It has been reported that they must each be recruited to the stimulated T-cell antigen receptor complex and the activated [28].”
Our response to minor comment 6:
Since small-molecule bands are always observed regardless reaction time of the protein synthesis, we guess it is not degradation by protease. They would be incomplete-length products or unfavorable products. We added the following sentences in Section 3.3 (lines 316-317).
“The small molecule bands observed in some lanes are considered to be incomplete length products.”
Reviewer 3 Report
In the manuscript "Characterization of phosphorylation status and kinase activity of Src tyrosine kinase family expressed in cell-based and cell-free protein expression systems" the authors address the problem of proper post-translational modifications in recombinant proteins, produced using different expression systems. The authors produce Src family kinases (SFKs) in various cell-based and cell-free protein expression systems and use Phos-tag SDS-PAGE to analyze the phosphorylation status and activity of the kinases. This is an important study indicating which methods to choose in order to obtain particular recombinant SFKs at required quality to basic and applied research. I find the manuscript valuable and suitable for publication after addressing the issues and introducing the changes indicated below.
Major points:
1. Line 353-354: Reference to "data not shown". There should be no conclusion on the basis of data that is not shown, and no reference to such data. I believe it is possible to include the data in the supplement.
2. The reader is supposed to assume the figures contain representative results of the experiments repeated a certain number of times. However, there is no such information, and no indication how many independent repeats were applied. This is crucial information to assess the quality of the data.
3. The authors use Phos-tag SDS-PAGE to evaluate the kinase activity of the analyzed SFKs, without quantifying the bands. The quantification would allow the authors to make more reliable comparisons and benefit the study.
4. The phosphorylation status of SFKs produced with different methods is successfully analyzed by Phos-tag SDS-PAGE. However, the technique can only indicate that there are some phosphorylated species of the protein, without knowing which residues are phosphorylated. Application of other methods, for example MS, could give more specific data to compare different expression systems, and would be very valuable for future users.
Minor points:
1. Line 42-44: The authors write that E. coli expression system is useful mainly for prokaryotic protein production. However, this system is often and successfully used for eukaryotic proteins that don't require specific post-translational modifications to remain active.
2. The kinases Fyn and Lck failed to be produced in active form using all expression systems. The authors could discuss more in the text whether that is a general issue for those kinases, or a particular problem noticed just in this study.
Author Response
Correspondence to Reviewer 3:
We really appreciate your constructive comments. The following are our replies to the comments.
Our response to major comment 1
The data were added in Fig. S3. We added the “Fig. S3” in section 3.4 (lines 360).
Our response to major comment 2
Cell-free expression, GST-srctide phosphorylation assay, 293 expression, and co-expression in E.coli have been performed at least 3 times. Images cannot be quantified, and deviation cannot be shown, but the clearest image was selected. Fig.2A/Fig.4C and Fig.4C/Fig5 contain data of the same experiment conducted independently. In addition, Fig. S1(Raw image for Fig. 2) shows the experiment was performed multiple times.
As our reply, we added the following sentences in experimental section 2.4 (lines 186-189).
“The Phos-tag SDS-PAGE images shown in this study were selected from images from three or more independent experiments showing a clear banding pattern without distortion. The raw images were shown in supplementary figures.”
Our response to major comment 3
In the E.coli co-expression system (Fig.3B), the expressed SFKs and the substrate GST-srctide were so small detectable by WB, and it was difficult to quantify the amount of both kinase and substrate. Therefore, the experiment is not quantitative, so we didn’t mention the comparison of activity among samples. However, the co-expression method has the advantage to easily determine whether the expressed kinase is active or not without purifying them.
Similarly, in vitro GST-srctide phosphorylation assay (Fig. 2B and Fig.4B) was performed to determine whether the expressed kinases are active or not. This experiment was not quantitative because it involves factors such as protein yield or efficiency of IP. So, we couldn’t mention the comparison of activity among samples. However, this method has the advantage to easily determine whether the expressed kinase is active or not.
Our response to major comment 4
The features of Phos-tag SDS-PAGE are that the difference in phosphorylation states among samples is visualized as the banding pattern, and the number of phosphorylateed species existence in each sample is shown. These are advantageous over the MS analysis. We believe that Phos-tag SDS-PAGE achieves the purpose of this study to demonstrate the differences in phosphorylation status among expression systems. As you pointed out, it is important to identify phosphorylation of each Phos-tag band. Like the KD experiment (Fig2A and Fig.5), alanine scans on all potential phosphorylation sites enable identification. These are our future challenges.
Our response to minor comment 1
As our reply, we added the following phrase in Introduction lines 45-46.
“or eukaryotic proteins that don’t require post-translational modifications.”
Our response to minor comment 2
Lck and Fyn are activated in T-cells rather than 293. It has been reported that they must each be recruited to the stimulated T-cell antigen receptor complex and the activated. As our reply, Ref.28 is newly cited and the following sentence was added in Discussion (lines 406-408)
“Lck and Fyn, which were not active in 293 cells, are involved in T-cell antigen receptor signaling. It has been reported that they must each be recruited to the stimulated T-cell antigen receptor complex and the activated [28]. “was
Round 2
Reviewer 2 Report
The authors did not addressed my main concerns. In particular, they did not provide any specific phosphorylation/activation, a result the is absolutely necessary for the significance of the study.
Reviewer 3 Report
The authors responded to my comments and addressed them by making appropriate changes in the text and adding missing data to the supplementary file. In my opinion, the most important issues are now solved and the manuscript is improved. I recommend the manuscript for publication.